# Effect of Co-Inoculation with Growth-Promoting Bacteria and Arbuscular Mycorrhizae on Growth of *Persea americana* Seedlings Infected with *Phytophthora cinnamomi*

**DOI:** 10.3390/microorganisms12040721

**Published:** 2024-04-02

**Authors:** Richard Solórzano-Acosta, Marcia Toro, Doris Zúñiga-Dávila

**Affiliations:** 1Centro Experimental La Molina, Dirección de Supervisión y Monitoreo en las Estaciones Experimentales Agrarias, Instituto Nacional de Innovación Agraria (INIA), Av. La Molina N° 1981, Lima 15024, Peru; 2Laboratorio de Ecología Microbiana y Biotecnología, Departamento de Biología, Facultad de Ciencias, Universidad Nacional Agraria La Molina, Lima 15024, Peru; mtoro@lamolina.edu.pe; 3Centro de Ecología Aplicada, Instituto de Zoología y Ecología Tropical, Facultad de Ciencias, Universidad Central de Venezuela, Caracas 1041-A, Venezuela

**Keywords:** *Bacillus*, *Pseudomonas*, PGPB, antagonists, root rot

## Abstract

Avocado is one of the most in-demand fruits worldwide and the trend towards its sustainable production, regulated by international standards, is increasing. One of the most economically important diseases is root rot, caused by *Phythopthora cinnamomi*. Regarding this problem, antagonistic microorganism use is an interesting alternative due to their phytopathogen control efficiency. Therefore, the interaction of arbuscular mycorrhizal fungi of the phylum Glomeromycota, native to the Peruvian coast (GWI) and jungle (GFI), and avocado rhizospheric bacteria, *Bacillus subtilis* and *Pseudomonas putida*, was evaluated in terms of their biocontrol capacity against *P. cinnamomi* in the “Zutano” variety of avocado plants. The results showed that the GWI and *Bacillus subtilis* combination increased the root exploration surface by 466.36%. *P. putida* increased aerial biomass by 360.44% and *B. subtilis* increased root biomass by 433.85%. Likewise, *P. putida* rhizobacteria showed the highest nitrogen (24.60 mg ∙ g^−1^ DM) and sulfur (2.60 mg ∙ g^−1^ DM) concentrations at a foliar level. The combination of GWI and *Bacillus subtilis* was the treatment that presented the highest calcium (16.00 mg ∙ g^−1^ DM) and magnesium (8.80 mg ∙ g^−1^ DM) concentrations. The microorganisms’ multifunctionality reduced disease severity by 85 to 90% due to the interaction between mycorrhizae and rhizobacteria. In conclusion, the use of growth promoting microorganisms that are antagonistic to *P. cinnamomi* represents a potential strategy for sustainable management of avocado cultivation.

## 1. Introduction

Avocado is one of the most popular tropical fruits in the world due to its high nutritional value and pleasant flavor [1]. Various problems affect the avocado crop, which reduce its yield and, therefore, result in lower income for producers [2] due to poor soil and water resource management that lead to root rot disease [3].

Root rot caused by *Phytophthora cinnamomi* is the most destructive avocado disease worldwide [4]. Primary symptoms are shown in the roots, which take on a dark brown color, become brittle, are easily detached, lose salt selectivity, and get infected by vascular pathogens. Secondary symptoms begin with leaves yellowing, progressive defoliation, and general decay (epinasty), similar to water stress, despite having soil in optimal humidity conditions [5].

Available control methods can reduce this disease’s severity [6]. However, the current trend is to incorporate new technologies that reduce agrochemical use to obtain healthier products and, at the same time, open more demanding markets [7]. In this sense, rhizobacteria and mycorrhizae constitute important sources of potentially beneficial microorganisms with plant growth-promoting activity and antagonistic effects against phytopathogens [8,9].

Arbuscular mycorrhiza-forming fungi (AMF) belonging to the Glomeromycota phylum are a crucial part of rhizospheric microbial communities [10]. These microorganisms establish symbiotic relationships with plants, playing an essential role in their nutrition and development. Moreover, they offer additional benefits such as increasing plants’ tolerance to biotic stresses like phytopathogen control. As a result, the commercial use and mass production of these fungi have generated significant interest [11].

Mycorrhizae are known to enhance the resistance of plants to pathogen attacks [12], particularly those that affect the root. When fungi establish themselves before pathogens, they reduce the incidence of damage. The benefits of mycorrhizae include an increase in root system biomass and nutrient absorption, which is facilitated by the fungal external mycelium [13]. Additionally, mycorrhizal plants activate the expression of genes that are related to resistance to pathogens. This may be attributed to other mechanisms that mycorrhizae activate in plants [14].

Recent research indicates that rhizobacteria, which are naturally found in soils, have the potential to control the harmful effects of *P. cinnamomi*-related diseases [15,16]. The *Bacillus* and *Pseudomonas* strains are effective as biofertilizers and biocontrol agents in agriculture. These bacteria can suppress pathogenic microbes, promote plant growth, and facilitate the assimilation of nutrients, resulting in beneficial effects [17,18].

For these reasons, this study of interactions between growth-promoting microorganisms will allow for overcoming the negative effect of root rot caused by *P. cinnamomi* on avocado crops. In this sense, this research’s main objective is to evaluate the effects of Phylum Glomeromycota mycorrhizae and *B. subtilis* and *P. putida* rhizobacteria co-inoculation on severity reduction of *P. cinnamomi* in “Zutano” variety avocado plants. Likewise, this research aims to identify the response in terms of vegetative and root growth promotion in plants infested with the disease. Finally, it seeks to relate co-inoculation, plant growth response, and macronutrient accumulation at a foliar level.

## 2. Materials and Methods

### 2.1. Bacteria–Mycorrhiza Co-Inoculation Trial Design

A complete randomized design (CRD) with factorial arrangement was used in this experiment. Two factors, each with three levels, were tested. The factors were mycorrhizae (without mycorrhizae, coastal mycorrhizal native *Glomeromycota* fungi from a wetland (GWI), and jungle mycorrhizal native *Glomeromycota* fungi from a fallow field (GFI)) and PGP bacteria with antagonistic activity in vitro on *P. cinnamomi* (without bacteria, *B. subtilis* Bac F, and *P. putida* P3). The experiment generated a total of nine treatments, each with six replicates, and one plant per replicate. For this trial, the seeds were first germinated, then inoculated with bacterial promoter, and, after a week, with mycorrhizae, depending on the treatments. The inoculations were allowed to be established for 21 days after inoculation with mycorrhizae before the plants were exposed to the phytopathogen *P. cinnamomi*. Seed germination and a description of the substrate used for the trials are discussed below.

Premix^®^ Nº8 substrate was used (pH: 5.5, EC: 0.75 dS ∙ m**^−^**^1^, P: 25 ppm, and K: 100 ppm), which was moistened to field capacity and sterilized in an autoclave at 121 °C and 15 lbs pressure. The sterile substrate was added to 2 L plastic pots. One avocado var. Zutano seed was planted per pot, weighing approximately 50 g, which was irrigated with 100 mL of sterile water. The seeds, donated by the Avo Hass Perú company, were previously sterilized in 10% sodium hypochlorite for 10 min, and once rinsed, they were sown and allowed to germinate for ten days in a greenhouse with 20 °C average temperature and 60% humidity until seedlings were obtained before treatment with the inoculants.

### 2.2. Inoculating Seedlings with Bacterial Strains

The bacteria used in this research were isolated from the avocado rhizosphere and were selected for their ability to grow in saline conditions in vitro. This was the subject of another publication in which their effectiveness in attenuating salinity was tested. Strains were cultured in nutrient broth for 48 h at 28 °C. Then, 30 mL of the bacterial broth from each strain, with a 108 CFU ∙ mL**^−^**^1^ concentration, was applied to the plant neck. After 15 days of germination, this was performed in the rootlet’s influence area. This procedure was repeated a second time, 3 days after the first inoculation, to ensure that the bacteria were present and suitable for the plant rhizosphere before starting the treatments.

### 2.3. Arbuscular Mycorrhiza Selection, Propagation, and Inoculation

Two *Glomeromycota* fungi inocula isolated from Peru were used. One was obtained from rhizosphere soil from a fallow land in Pucallpa, Ucayali (*Glomeromycota* fallow inoculum—GFI), and the other from rhizosphere soil from *Sporobolus* sp. located in Pisco, Ica (*Glomeromycota* wetland inoculum—GWI), according to Castañeda et al. [19]. Both were reproduced in trap pots with *Brachiaria decumbens* in sterile sand, and watered with Long Ashton’s solution and a quarter dose of phosphorus (P) every 15 days [20].

Rhizosphere samples were taken to quantify spores and arbuscular mycorrhizae (AM) colonization of rootlets [21] in each inoculum. Similar spore numbers were obtained in GFI and GWI to inoculate avocado plants in the experiment. The GFI inoculum consisted of native species, mainly *R. intraradices* fungus and *Acaulospora*, *Gigaspora*, and *Archaeospora* genera, of which 3726 spores per mL per seedling were added from colonized rootlets (70%). In the GWI inoculum, *R. intraradices* fungus predominated, of which 3400 spores per mL per seedling were applied from colonized rootlets (90%). In both cases, inocula were applied to the base of the germinated seeds, then spread around, and the substrate was uncovered slightly with a sterile spatula to expose the rootlets without hurting them.

### 2.4. Propagation and Inoculation of P. cinnamomi

Propagation was carried out on carnation petals, which were previously disinfected in 1% sodium hypochlorite for 60 s, and then rinsed with sterile water. Petals were placed in a 250 mL sterile flask with 100 mL of sterile mineral water. Then, agar discs containing *P. cinnamomi* were placed on the petals, allowing the pathogen to colonize them. Afterward, inoculation was performed (Figure 1). Infection of seedlings with *P. cinnamomi* was carried out by applying 100 mL of zoospore solution at 105 zoospores ∙ mL**^−^**^1^ concentration around the avocado seedling neck on two occasions spaced one week apart. The *P. cinnamomi* strain was identified by the Laboratorio Agrícola Biaster SAC company in Ica. Pathogen inoculation was carried out two weeks after the last inoculation with mycorrhizae to promote their establishment. It should be mentioned that prior inoculation is crucial because, as this is an irreversible disease, a preventive or mitigating effect is sought before infection.

### 2.5. Greenhouse Environmental Conditions

During the experiment, the highest temperatures recorded ranged from 24.51 to 27.43 °C, while the lowest temperatures ranged from 18.91 to 20.37 °C. The relative humidity varied between 59.9 and 61.92%. These measurements were taken by the meteorological station situated inside the greenhouse of the Microbial Ecology and Biotechnology Laboratory at the National Agrarian University La Molina.

### 2.6. Determination of Harvest, Growth, and Nutrition Characteristics

The experiment lasted for 20 weeks and involved growing plants from germination to harvesting for nutrient analysis. The germination process took 2 weeks, followed by inoculation and establishment of PGP bacteria and arbuscular mycorrhizae for 4 weeks, infection with the pathogen for 2 weeks, and finally, plant growth for 12 weeks. After this time, various characteristics of plant growth, such as plant height (cm), root length (cm), and number of leaves, were measured. Fresh and dry aerial biomass (g), fresh and dry root biomass (g), root moisture (g), and root length/root dry weight ratio were also evaluated as an index of response to the pathogen that causes root damage. The dry biomass was determined by drying aerial and root parts in an oven for two weeks at 40 °C. Root humidity was calculated by finding the difference between fresh and dry biomass, which indicates the root tissue water content and root health state.

### 2.7. P. cinnamomi Infection Severity

The severity of *P. cinnamomi* infection in avocado seedlings that were previously inoculated with *Pseudomonas* sp. and *Bacillus* sp. strains was determined by using the visual scale proposed by Ramírez and Morales (2020). The scale is based on external symptoms and was validated with the inoculum amount through regression analysis.

### 2.8. Statistical Analysis

The statistical analysis of the experiment was conducted using IBM’s Statistic Package for Social Sciences (SPSS) program, version 26. Data collected in each experiment were analyzed using a complete randomized design (CRD) with factorial arrangement through test F. The tested factor’s main effects were calculated, and in case of significance, the treatments were compared using Duncan’s test to determine differences between bacterial strains and mycorrhizae. A probability of alpha error of less than 5% was considered significant.

## 3. Results

### 3.1. Effect of Bacteria–Mycorrhiza Co-Inoculation on P. Americana var. Zutano Seedlings’ Growth Parameters after Infection with P. cinnamomi

Mycorrhizae show general activity affecting avocado root length and the relationship between root length and root dry weight, which results in greater root efficiency by generating a greater exploration surface concerning the biomass generated (Figure 2, Figure 3 and Figure 4). They also reduce the severity of the disease (Table 1, Figure 5). In all cases, the presence of mycorrhizae in the roots was verified (Figure 6). Additionally, *B. subtilis* and *P. putida* bacteria promote plant growth by positively affecting various growth parameters such as plant height, root length, number of leaves, fresh and dry biomass, and root humidity. *B. subtilis* stands out for its positive impact on root humidity (Table 2). However, co-inoculating mycorrhizae with antagonistic bacteria such as *P. putida* and *B. subtilis* can decrease plant height (Figure 2). There is a positive synergistic effect when coastal mycorrhizae and *B. subtilis* are combined, promoting root length (Figure 3). The same is seen with jungle mycorrhizae and *B. subtilis* for the number of leaves. Inoculating *P. putida* favors aerial biomass production, while *B. subtilis* favors overall avocado seedling growth, especially the accumulation of aerial and root biomass (Table 2).

In Figure 5, it is depicted that the presence of mycorrhizae enhances the plants’ appearance, making them more vigorous despite being infected by *P. cinnamomi*. Moreover, there were no decay symptoms observed in either inoculated type. The jungle mycorrhiza consortium influences pathogen resistance, and in the presence of PGP bacteria, both result in a stimulating effect. However, there was no uniform response among plants, and therefore plant height may not necessarily correspond in direct proportion to dry biomass and root growth characteristics, as shown in Figure 2 and Table 2.

Avocado seedlings’ root structure is less affected by *P. cinnamomi* in the presence of mycorrhizae and the PGP bacteria *B. subtilis* and *P. putida*, as demonstrated in Figure 7.

### 3.2. Principal Component Analysis (PCA)

A PCA was performed on the biometric variables evaluated. In both inoculant treatments, with rhizobacteria and mycorrhizae, an explained variance of more than 77% was reached with the first two components. A biplot graph including variables together with observations is shown in Figure 8 and Figure 9. First, with respect to the variables, the existence of a higher correlation between root length and root exploration is observed, expressed in the relationship between root length and root biomass. On the other hand, there are high correlations between root fresh and dry weight, fresh and dry weight of the aerial part, number of leaves, plant height, and root moisture. Similarly, it was observed that the variables that contributed most to the PCA in the rhizobacteria and mycorrhizae treatments were root length and root moisture.

Plants not inoculated with rhizobacteria were placed on the negative side of the first component (PC1) and plants inoculated with *B. subtilis* and *P. putida* on the positive side. Plants inoculated with both rhizobacteria were associated with a greater increase in aerial and root biomass. On the other hand, in the case of plants inoculated with mycorrhizae, no clear trends in their behavior were observed. On the other hand, plants inoculated with *B. subtilis* and coastal mycorrhizae (GWI), which are positioned in all quadrants of the biplot, do not show a clear behavior in terms of indicators.

### 3.3. Effect of Bacteria–Mycorrhiza Co-Inoculation on P. americana var. Zutano Macroelement Content after Infection with P. cinnamomi

Plant leaves infected with *P. cinnamomi* and inoculated with bacteria and mycorrhizae were analyzed. It was found that mycorrhizae do not have a significant impact on macronutrient accumulation. On the other hand, bacteria do improve macronutrient accumulation. *P. putida* improves nitrogen, potassium, and sulfur accumulation and *B. subtilis* increases calcium and magnesium accumulation (Table 3).

Inoculation with *P. putida* promotes nitrogen and sulfur accumulation. Additionally, there is a positive interaction between the GWI coastal mycorrhizae and the antagonist bacteria *B. subtilis* and *P. putida*, especially when compared to the GFI jungle mycorrhizae. This interaction results in increased absorption of calcium and magnesium by *B. subtilis* and increased potassium content caused by *P. putida*, as shown in Table 3. 

## 4. Discussion

As observed, mycorrhizae increase avocado root length and root length per root dry weight ratio in plants infected by *P. cinnamomi*, as well as the number of leaves (Table 2). Few references show mycorrhizal inoculation effects on avocados, but all of them agree that there is a stimulating growth effect in general and specifically on the root [9,22,23,24]. Regarding dry weight, Viera et al. [22] reported that mycorrhizal fungi contributed to a greater amount of dry matter in avocado seedlings in greenhouse conditions by up to 44% more compared to the uninoculated control, perhaps because they work in *P. cinnamomi* infection conditions, unlike the present investigation where the dry matter remained statistically similar to the control.

The interaction between GWI and B. subtilis resulted in the greatest root length and exploration increases and the greatest calcium uptake. Calcium is an essential nutrient for cell wall and membrane stability against *P. cinnamomi*’s hyphae penetration [25]. It is also involved in early defense signaling against pathogen attacks [25]. Although mycorrhizae have been associated with reduced calcium uptake due to endodermis lignification and suberification [26], this problem can be overcome by *B. subtilis*’ synergistic effects. *B. subtilis*’ main effect on plants is a root hair density increase, stimulated by N-acyl-L-homoserine lactones (AHLs), cyclodipeptides (CDPs), and volatile organic compounds [27]. However, the combined effect with GWI resulted in greater root biostimulation rather than B. subtilis’ independent effect. This is because mycorrhizae increase Zn uptake [26]. Zn is a precursor nutrient for biosynthesis of auxin, a hormone whose main physiological effect is root branching [28]. In this sense, higher root hair density increases calcium uptake and reduces the severity of root rot caused by *P. cinnamomi*.

No reports were found regarding the effect that mycorrhizae have under *P. cinnamomi* infection conditions, specifically on *P. americana*. However, Lara et al. [29] identified mycorrhizal species present in root rot-infected plants, concluding that despite the infection mycorrhizae are present and their diversity is considerable. However, in other species such as oak, it was reported that the presence of *P. cinnamomi* altered relationships between and the abundance of ectomycorrhizae [30]. Furthermore, in oak, the appearance of mycorrhizae in the substrate improved acorn germination in the presence of *P. cinnamomic* [31]. As proposed by Shu et al. [9], entire soil biotic communities, including mycorrhizae, enhance plant resistance to *P. cinnamomi* and may moderate *P. cinnamomi*-induced mortality.

It was discovered that the bacteria *B. subtilis* and *P. putida* can enhance the growth of avocado plants by positively influencing various growth parameters such as plant height, root length, number of leaves, and fresh and dry biomass. This effect is particularly noticeable when the avocado plants are infected with *P. cinnamomi*, and *B. subtilis* is more effective in such conditions as compared to *P. putida* (Table 2). In situations of biotic and abiotic stress, such as pathogen infection, beneficial effects on plant growth are reported due to the production of enzymes and indoles and nutrient solubilization [32,33]. There is a lot of important information available on bacterial evidence for controlling root rot in avocados. Several studies [8,15,34] have shown that *Bacillus subtilis* isolated from the avocado rhizosphere can reduce up to 25% of the mycelial growth of *P. cinnamomi*. Even a systematic review has pointed out that *Bacillus* is very promising for controlling and inhibiting several species of *Phytophthora* [35]. Another study has demonstrated the potential for biological control of avocado root rot with *P. fluorescens* [7]. This bacterium also induces root and apical growth of avocado, even under *P. cinnamomi* infection conditions [17]. Such activity is associated with the production of antibiotics. 

Gil et al. [36] found that co-inoculation with *Pseudomonas* sp. and *Glomus fasciculatum* significantly improves avocado seedlings’ growth. Seedlings’ height was found to improve after growth-promoting bacteria application, with the greatest increase seen by using the case of *B. subtillis*. However, similar statistically significant improvements were observed when GWI and *P. putida* were combined, indicating a positive interaction between the two microorganisms. This interaction also increases fresh root biomass, root humidity, and dry aerial biomass. In cases where seedlings were infected by *P. cinnamomi*, the presence of mycorrhizae sustained the effect of the promoting bacteria. However, the number of leaves and fresh aerial biomass were not significantly affected by the interaction with mycorrhizae. In these cases, the favorable response was greater when only growth-promoting bacteria were present. In terms of dry root biomass accumulation, the presence of any of the microorganisms or a bacteria–mycorrhiza association improved this characteristic. However, a greater effect was observed when only *B. subtillis* was present. The severity of the infection was also reduced when promoter bacteria and mycorrhizae were inoculated together, as compared to when only one type of microorganism was present or when both were absent. Overall, a positive interaction between promoting bacteria and mycorrhizae was observed in promoting avocado growth. However, this interaction may be specific to certain species and crops. Further ecological studies are needed to clarify these situations in greater detail.

*P. cinnamomi* infection depends on the photosynthates’ concentration in the root surface apoplast, and their accumulation depends on the root cell’s plasma membrane permeability [25]. At the nutritional level, there are two causes underlying the plant’s susceptibility. First, potassium deficiency and nitrogen excess increase soluble sugar accumulation in cells; i.e., there is a greater substrate amount for the pathogen’s mycelial development [37]. Secondly, calcium, boron, and zinc deficiencies increase plasma membrane permeability; that is, there is an increased sugar availability for *Phytophthora* [37]. *Pseudomonas putida* interaction with both mycorrhizae resulted in a treatment that reached a higher concentration of foliar potassium, which justifies a smaller amount of disease development, demonstrated by a greater plant height increase and a higher aerial biomass formation. Likewise, the interaction between GWI and *B. subtilis* and the individual effect of *Bacillus* resulted in the greatest leaf calcium increases, the inhibition of disease development, and the promotion of major growth in terms of root length and exploration.

Accumulation of macroelements in avocados is significantly influenced by inoculating them with *B. subtilis* and *P. putida*. Similar effects have been observed in other species such as pearl millet, where the application of *Bacillus* sp. endophytes increases N, P, and K accumulation in sprouts. In addition, studies have shown that co-inoculating with *Bacillus* and *Pseudomonas* genera can also be beneficial. For instance, He et al. [38] found that *P. putida* significantly increased macronutrient and micronutrient content in tomato fruits when co-inoculated with *Bacillus*.

It has been mentioned that co-inoculation in stevia leads to a synergistic effect in nitrogen, phosphorus, and potassium accumulation, resulting in increased nutrient content. Therefore, an appropriate combination of mycorrhizal fungi and PGPR as biotic inducers can enhance plant growth and nutrient content, as stated by Vafadar et al. [39].

Evidence supporting the use of growth-promoting microorganisms to increase the nutrient content of crops, including roots, leaves, and fruits, is still being researched. The effectiveness of inoculation varies depending on the specific microorganism used, as well as the type and amount of nutrient that is being targeted for accumulation through microbial stimulation [40,41,42].

## 5. Conclusions

Under the infection conditions caused by *P. cinnamomi*, a positive synergistic effect was observed between coastal mycorrhizae and *B. subtilis* on root length improvement in avocado rootstock seedlings of var. Zutano. Similarly, when jungle mycorrhizae and *B. subtilis* were inoculated together, a significant increase in leaf number was observed.

Both *P. putida* and *B. subtilis* have been found to have significant effects on avocado seedlings’ growth, especially in terms of biomass accumulation, in the absence of mycorrhizae. In terms of nutrient uptake, *P. putida* inoculation has been found to enhance nitrogen and sulfur accumulation, even without mycorrhizae interaction.

GWI coastal mycorrhizae positively interact with antagonist bacteria *B. subtilis* and *P. putida*, leading to increased calcium, magnesium, and potassium absorption compared to GFI jungle mycorrhizae.

Regarding *P. cinnamomi* damage, indirect parameters such as fresh and dry weight are indicators of root vigor and morphology associated with pathogen damage, in which case the presence of *P putida* and *B. subtillis* increases fresh and dry root biomass independently of arbuscular mycorrhizae association. Similarly, the root length/root dry weight ratio indicates greater production of secondary roots due to dry matter accumulation, so the higher the ratio, the greater the pathogen damage attenuation, which was demonstrated with the joint use of GWI and *B. subtillis* that obtained the highest index with 15.39, surpassing other treatments. Furthermore, according to the qualitative severity scale, the presence of growth-promoting bacteria and mycorrhizae appears to reduce damage. Therefore, it is concluded that the co-inoculation of PGP bacteria and arbuscular mycorrhizae promotes the growth of *P. Americana* seedlings infected with *P. cinnamomi*.

## Figures and Tables

**Figure 1 microorganisms-12-00721-f001:**
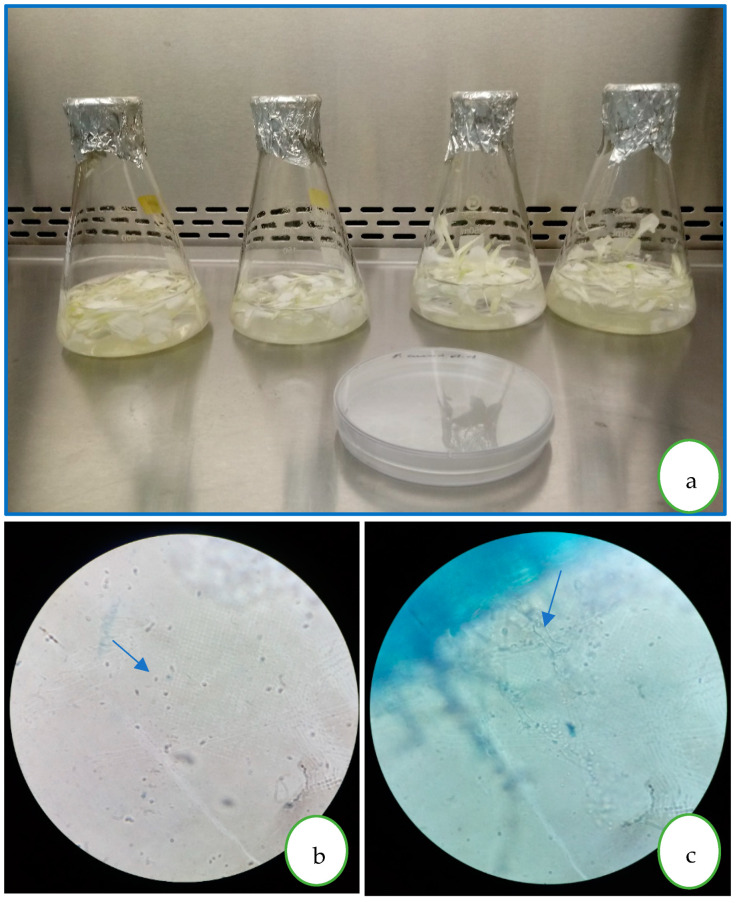
(**a**) *P. cinnamomi* inoculum preparation in mineral water and carnation petals; (**b**) zoospores (1000×); (**c**) mycelium and sporangia on carnation petals.

**Figure 2 microorganisms-12-00721-f002:**
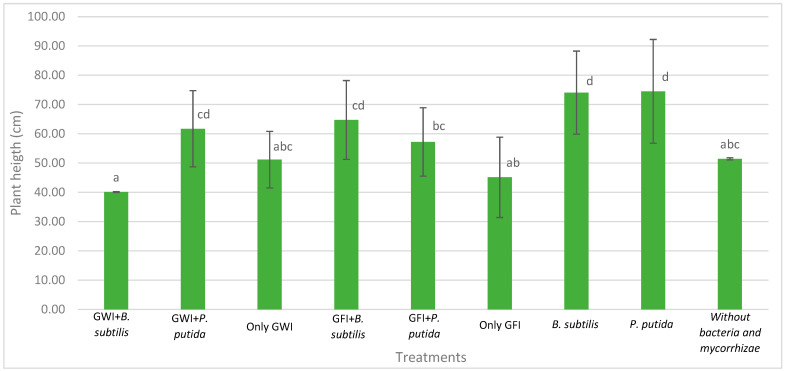
Effects of mycorrhizae and PGP bacteria on *P. americana* var. Zutano seedling height after infection with *P. cinnamomi*. Averages with the same letter in a column are statistically similar according to Duncan’s test at a 95% confidence level. Bars indicate standard deviation.

**Figure 3 microorganisms-12-00721-f003:**
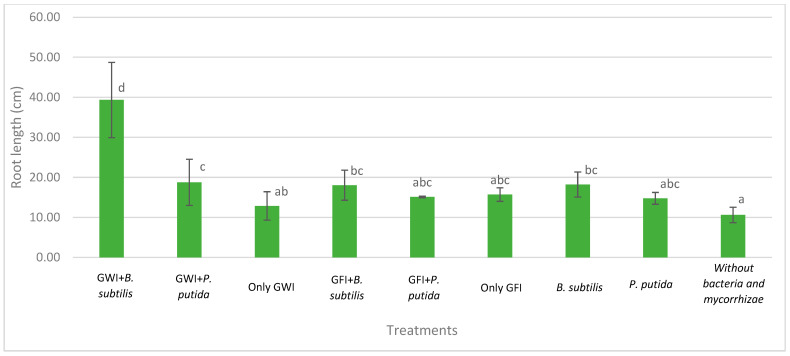
Effects of mycorrhizae and PGP bacteria on *P. americana* var. Zutano seedling root length after infection with *P. cinnamomi*. Averages with the same letter in a column are statistically similar according to Duncan’s test at a 95% confidence level. Bars indicate standard deviation.

**Figure 4 microorganisms-12-00721-f004:**
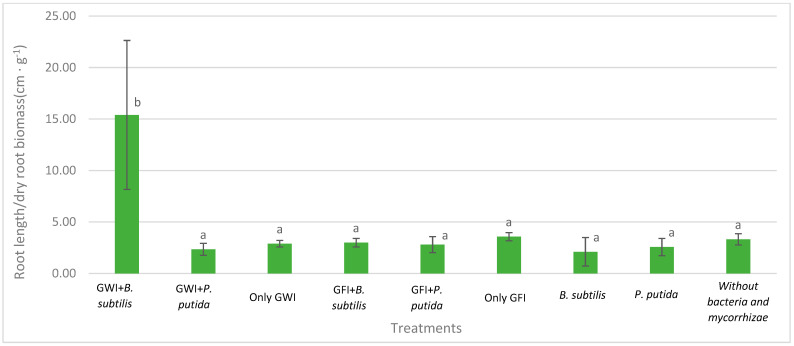
Effects of mycorrhizae and PGP bacteria on *P. americana* var. Zutano root length/root dry biomass after infection with *P. cinnamomi*. Averages with the same letter in a column are statistically similar according to Duncan’s test at a 95% confidence level. Bars indicate standard deviation.

**Figure 5 microorganisms-12-00721-f005:**
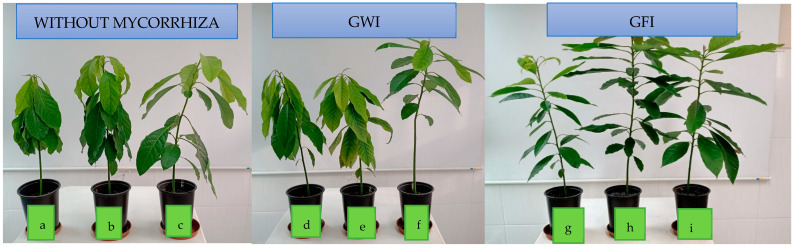
*P. Americana* var. Zutano appearance after co-inoculation with bacteria/mycorrhizae and infection with *P. cinnamomi*: (**a**) without mycorrhizae + without bacteria; (**b**) without mycorrhizae + *B. subtilis*; (**c**) without mycorrhizae + *P. putida*; (**d**) GWI + without bacteria; (**e**) GWI + *B. subtilis*; (**f**) GWI + *P. putida*; (**g**) GFI + without bacteria; (**h**) GFI + *B. subtilis*; (**i**) GFI + *P. putida*.

**Figure 6 microorganisms-12-00721-f006:**
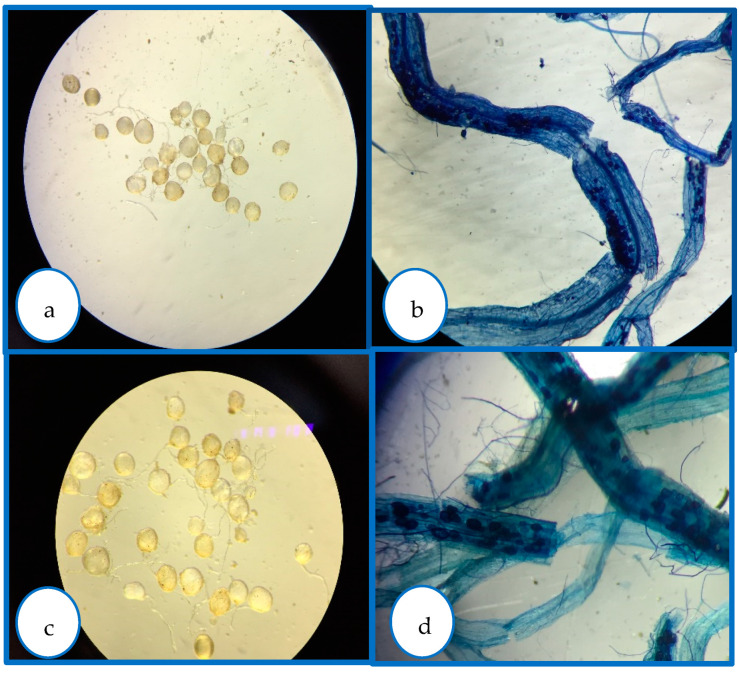
To confirm the presence of inoculated mycorrhizae spores in avocado roots, the spores were stained with trypan blue. This figure displays the differences in spore morphology and root colonization: (**a**) mycorrhizal roots; (**b**) mycorrhizal spores; (**c**) GWI (*Glomus intraradices*); (**d**) GFI (consortium).

**Figure 7 microorganisms-12-00721-f007:**
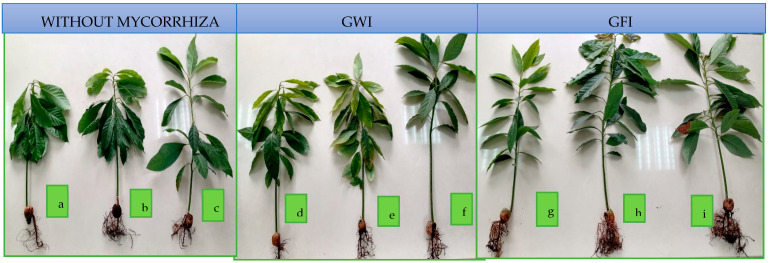
Effects of mycorrhizae and PGP bacteria on P. *americana* var. Zutano seedling roots after infection with *P. cinnamomi*: (**a**) without mycorrhizae + without bacteria; (**b**) without mycorrhizae + *B. subtilis*; (**c**) without mycorrhizae + *P. putida*; (**d**) GWI + without bacteria; (**e**) GWI + *B. subtilis*; (**f**) GWI + *P. putida*; (**g**) GFI + without bacteria; (**h**) GFI + *B. subtilis*; (**i**) GFI + *P. putida*.

**Figure 8 microorganisms-12-00721-f008:**
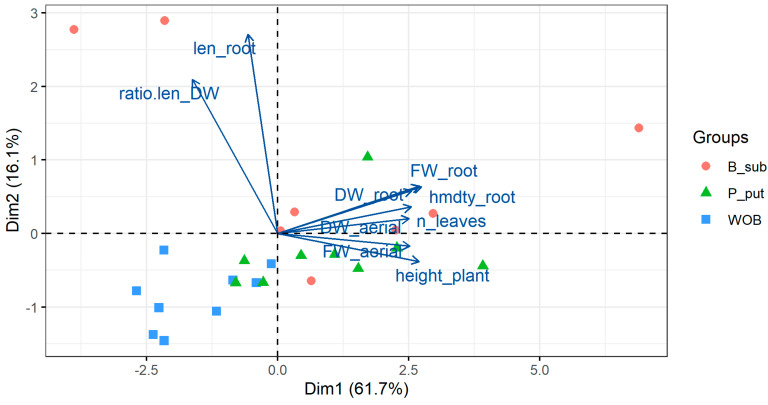
PCA biplot graph (variables together with observations) considering all data and the results in plants inoculated with *B. subtilis* and *P. putida* rhizobacteria.

**Figure 9 microorganisms-12-00721-f009:**
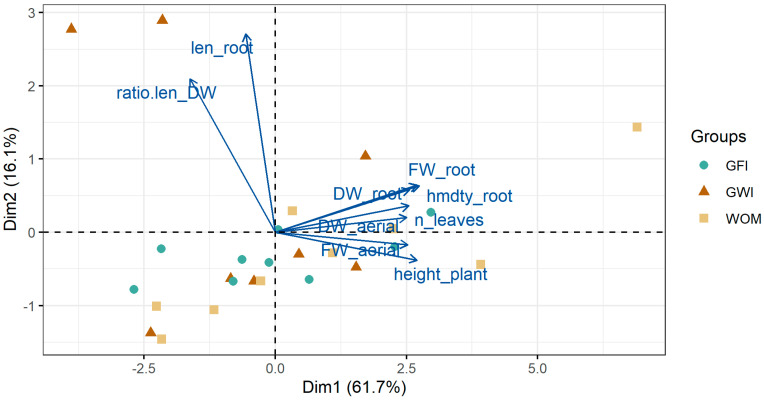
PCA biplot graph (variables together with observations) considering all data and the results in plants inoculated with coastal mycorrhizae (GWI) and jungle mycorrhizae (GFI).

**Table 1 microorganisms-12-00721-t001:** The severity of *Phytophthora cinnamomi* disease in avocado was evaluated based on different treatments.

Treatment	Scale Value	Outbreak Appearance and Disease Symptom	Root Appearance
GFI/*B. subtilis*	1	Visible disease symptoms. General yellowing of leaves.	Diseased rootlets between 10 y 15%
GFI/*P. putida*	1	Visible disease symptoms. General yellowing of leaves.	Diseased rootlets between 10 y 15%
GFI/without bacteria	1	Visible disease symptoms. General yellowing of leaves.	Diseased rootlets between 10 y 15%
GWI/*B. subtilis*	1	Visible disease symptoms. General yellowing of leaves.	Diseased rootlets between 10 y 15%
GWI/*P. putida*	1	Visible disease symptoms. General yellowing of leaves.	Diseased rootlets between 10 y 15%
GWI/without bacteria	3	Generalized chlorosis, wilting, and defoliation.	Diseased rootlets > 70.1%.
Without mycorrhiza/*B. subtilis*	2	Generalized yellowing of leaves, stunted growth, and mild wilting.	Diseased rootlets between 15,1 y 70%
Without mycorrhiza/*P. putida*	2	Generalized yellowing of leaves, stunted growth, and mild wilting.	Diseased rootlets between 15,1 y 70%
Without mycorrhiza/without bacteria	3	Generalized chlorosis, wilting, and defoliation.	Diseased rootlets > 70.1%.

**Table 2 microorganisms-12-00721-t002:** Bacteria–mycorrhiza co-inoculation effect on *Persea americana* var. Zutano seedlings’ growth parameters after infection with *P. cinnamomi*.

Treatment	Number of Leaves	Fresh Aerial Biomass (g)	Dry Aerial Biomass (g)	Fresh Root Biomass (g)	Dry Root Biomass (g)	Root Humidity (g)	Root Length/Root Weight
	Media	Media	Media	Media	Media	Media	Media
GWI/*B. subtilis*	16.66 ± 2.07 ^ab^	31.57 ± 7.97 ^a^	16.59 ± 3.98 ^ab^	9.13 ± 2.94 ^a^	3.18 ± 1.62 ^a^	5.94 ± 1.32 ^ab^	15.39 ± 7.23 ^b^
GWI/*P. putida*	24.66 ± 5.96 ^bc^	72.42 ± 10.04 ^b^	33.49 ± 6.31 ^cd^	21.09 ± 1.7 ^b^	7.92 ± 0.49 ^a^	13.16 ± 1.2 ^d^	2.34 ± 0.58 ^a^
GWI/without bacteria	17.33 ± 4.41 ^ab^	44.88 ± 15.41 ^a^	18.82 ± 6.41 ^ab^	10.45 ± 2.53 ^a^	4.50 ± 1.38 ^a^	5.95 ± 1.19 ^ab^	2.89 ± 0.32 ^a^
GFI/*B. subtilis*	35.66 ± 6.95 ^d^	82.20 ± 18.31 ^b^	35.71 ± 8.12 ^cd^	15.05 ± 4.94 ^ab^	6.18 ± 1.79 ^a^	8.87 ± 3.15 ^abcd^	2.98 ± 0.42 ^a^
GFI/*P. putida*	24.00 ± 4.65 ^bc^	66.03 ± 23.51 ^b^	26.71 ± 9.7 ^bc^	16.00 ± 5.66 ^ab^	5.76 ± 1.64 ^a^	10.24 ± 4.03 ^bcd^	2.80 ± 0.77 ^a^
GFI/without bacteria	18.00 ± 8.53 ^ab^	29.95 ± 11.73 ^a^	11.37 ± 3.82 ^a^	11.59 ± 3.33 ^a^	4.45 ± 0.72 ^a^	7.14 ± 2.67 ^abc^	3.57 ± 0.39 ^a^
Without mycorrhiza/*B.subtilis*	31.00 ± 13.18 ^cd^	74.21 ± 11.47 ^b^	35.48 ± 7.93 ^cd^	32.66 ± 19.1 ^c^	13.97 ± 10.42 ^b^	18.69 ± 8.69 ^e^	2.09 ± 1.38 ^a^
Without mycorrhiza/*P. putida*	27.00 ± 5.37 ^c^	83.47 ± 22.11 ^b^	41.45 ± 15.13 ^d^	17.19 ± 5.02 ^ab^	6.31 ± 2.03 ^a^	10.88 ± 3.08 ^cd^	2.56 ± 0.84 ^a^
Without mycorrhiza/without bacteria	15.00 ± 4.98 ^a^	34.91 ± 12.33 ^a^	11.50 ± 8.04 ^a^	7.85 ± 0.8 ^a^	3.22 ± 0.37 ^a^	5.30 ± 0.62 ^a^	3.30 ± 0.54 ^a^

Averages with the same letter in a column are statistically similar according to Duncan’s test at a 95% confidence level.

**Table 3 microorganisms-12-00721-t003:** Bacteria–mycorrhiza co-inoculation effects on *P. americana* var. Zutano seedling macronutrient absorption after infection with *P. cinnamomi*.

Treatment	N	P	K	Ca	Mg	S
	mg g^−1^ DM					
GWI/*B. subtilis*	19.0 ± 0.8 ^a^	1.2 ± 0.21 ^a^	14.2 ± 0.1 ^d^	16.0 ± 0.1 ^i^	8.8 ± 0.1 ^f^	1.8 ± 0.1 ^bc^
GWI/*P. putida*	21.8 ± 0.1 ^f^	1.4 ± 0.22 ^a^	17.7 ± 0.3 ^i^	12.6 ± 0.2 ^f^	6.9 ± 0.1 ^c^	2.1 ± 0.1 ^d^
GWI/without bacteria	20.2 ± 0.2 ^b^	1.4 ± 0.29 ^a^	13.2 ± 0.3 ^a^	9.2 ± 0.1 ^a^	5.5 ± 0.2 ^a^	1.9 ± 0.1 ^c^
GFI/*B. subtilis*	20.4 ± 0.1 ^c^	1.3 ± 0.11 ^a^	13.7 ± 0.1 ^b^	10.8 ± 0.1 ^c^	7.0 ± 0.2 ^c^	1.9 ± 0.1 ^c^
GFI/*P. putida*	21.3 ± 0.3 ^e^	1.2 ± 0.10 ^a^	16.5 ± 0.2 ^h^	11.9 ± 0.2 ^e^	6.5 ± 0.2 ^b^	1.6 ± 0.1 ^a^
GFI/Sin bacteria	19.0 ± 0.7 ^a^	1.2 ± 0.21 ^a^	15.5 ± 0.1 ^g^	11.0 ± 0.1 ^d^	6.5 ± 0.2 ^b^	1.7 ± 0.1 ^ab^
Without mycorrhiza/*B. subtilis*	23.8 ± 0.5 ^g^	1.4 ± 0.09 ^a^	15.1 ± 0.1 ^f^	13.9 ± 0.2 ^g^	8.3 ± 0.2 ^e^	2.2 ± 0.1 ^d^
Without mycorrhiza/*P. putida*	24.6 ± 0.2 ^h^	1.4 ± 0.18 ^a^	14.0 ± 0.1 ^c^	15.6 ± 0.1 ^h^	7.5 ± 0.1 ^d^	2.6 ± 0.2 ^e^
Without mycorrhiza/without bacteria	2.07 ± 0.1 ^d^	1.4 ± 0.08 ^a^	14.7 ± 0.4 ^e^	9.6 ± 0.2 ^b^	6.4 ± 0.2 ^b^	2.2 ± 0.1 ^d^

Averages with the same letter in a column are statistically similar according to Duncan’s test at a 95% confidence level.

## Data Availability

The data presented in this study are available on request from the corresponding author.

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
