# Peer review of "Effect of Co-Inoculation with Growth-Promoting Bacteria and Arbuscular Mycorrhizae on Growth of Persea americana Seedlings Infected with Phytophthora cinnamomi"

_microorganisms, 2024, doi:10.3390/microorganisms12040721_

Round 1

Reviewer 1 Report

Comments and Suggestions for Authors

The research evaluates a co-inoculation strategy using bacteria and mycorrhiza to control Phytophthora cinnamomi, the pathogen causing root rot in avocados. The study finds that this strategy enhances plant growth parameters and nutrient absorption while antagonizing the pathogen.

  • The methodology appears to be well designed, focusing on evaluating the efficacy of co-inoculation with bacteria and mycorrhiza in controlling P. cinnamomi.
  • A complete randomized design with a factorial arrangement was used. Two factors each with 3 levels were tested.
  • Details on the cultivation conditions, replication, and statistical analysis methods were described to ensure the reliability and reproducibility of the results.
    • The results suggest that the co-inoculation strategy positively impacts avocado plant growth parameters such as root length and moisture content, despite P. cinnamomi infection. This finding is promising and aligns with the objectives of sustainable agriculture by reducing reliance on chemical inputs.
    • The increased absorption of essential nutrients like potassium, calcium, and magnesium due to the presence of plant growth-promoting bacteria is another significant result. This indicates the potential of the co-inoculation strategy to enhance nutrient uptake and overall plant health.
    • The antagonistic effect observed on P. cinnamomi by both mycorrhiza and plant growth-promoting bacteria (e.g., B. subtilis and P. putida) is a crucial finding. However, the degree of antagonism and its mechanism need further elucidation for a comprehensive understanding of the biocontrol mechanism.
    • The study provides valuable insights into the potential of biological control methods in managing avocado root rot, a significant threat to avocado cultivation globally. Further research should focus on optimization and real-world applicability.
      It is a well written paper based on a well designed experimental protocol. Results are promising for agriculture .Bibliography is up to date.
      My suggestion is to ACCEPt and publish it in its present form.

Author Response

Reviewer 1: thank you for your comments, we trust we have done a good job with sobriety and accuracy.

Reviewer 2 Report

Comments and Suggestions for Authors

The presented manuscript is devoted to the effect of arbuscular mycorrhizal fungi in combination with rhizobacteria on the growth of an avocado plant infected with Phytophthora cinnamomi. The study conducted by the authors provides convincing evidence of the positive synergistic effect of mycorrhizal fungi and rhizobacteria on an infected avocado plant. In my opinion, this is a well-written manuscript; it has a good logical structure; all conclusions are supported by the data. I read it with interest and joy. In general, it can be published as is, without further modification. However, I suggest that authors re-read the manuscript with fresh eyes. Additionally, include not only the means but also the standard deviations in all the Tables and superscript all letters that refer to statistical significance; check the legend in Figure 4.

Comments on the Quality of English Language

There are some minor stylistic issues; therefore, I suggest that authors reread the manuscript with fresh eyes.

Author Response

Reviewer 2: thank you for your comments, we think that in the field the conditions may vary and the response may be affected; however, it is typical of this type of trials the treatment by scaling up, in which case this is a phase prior to field experimentation, although we have data on this, these will be part of another article. The summary has been rewritten indicating the improvements or differences in percentages. The relevance and novelty of the manuscript is indicated as suggested. Six replicates per treatment were made and one plant is a replicate or observation. Some mechanisms are added and commented as suggested but we are careful not to speculate or go beyond what this experiment can demonstrate. The mineral composition is reported in concentration and not in percentage. In addition, PCA was performed as suggested. 

Reviewer 3 Report

Comments and Suggestions for Authors

The research paper submitted to the journal fits within the general scope of Microrganisms MDPI. The experiment was well conducted and cover an interesting topic aboout PGPR and AMF in other words microbial biostimulants. In particular the effect of microbial biostimulants on the morphological and mineral composition responses on an important fruit trees such as avocado.

The major drawback of the trial is that the experiment is carried out in pots and short experiment. What will happen if the experiment was carried out under open field condistions.

The abstract should be re-written by indicating in percentage the differences between treatments.

At the end of the introduction the authors should clearly report their hypothesis and how their work is novel in comparison to previous published papers.

How many plant per replicates?

The discussion section is very weak it is not enough to report whether your results are in line or not with previous papers but why what was the mechanisms behind these differences.

the mineral composition should be reported in concentration and not in percentage.

A PCA is required at the end of the results

Author Response

Reviewer 3: Thank you for your suggestions. Accordingly, the document and the style were revised to find options for improvement. The deviations in the tabulated data were added. 

Reviewer 4 Report

Comments and Suggestions for Authors

Interesting article on biological control of avocado crops. Increasingly higher standards imposed in connection with the limitation of chemical agents require the development of biological methods of plant protection. Root rot caused by Phytophthora cinnamomi is quite a serious threat to avocado crops. This work concerns the possibility of using microorganisms (PGP) to combat phytopathogens. A strategy of simultaneous inoculation of bacteria and mycorrhizas was evaluated in the biocontrol of the avocado rootstock P. cinnamomi var. Zutan. The authors studied mycorrhizal bacteria and PGP showing in vitro antagonistic activity towards P. cinnamomi. The results obtained by the authors indicate that the presence of mycorrhizas improves the growth parameters of plants, making them stronger despite P. cinnamomi infection. Mycorrhizas and bacteria such as B. subtilis and P. putida act antagonistically on P. cinnamomi.
The materials and methods were described in detail. Statistical analysis was performed by selecting appropriate tests and statistically significant differences were demonstrated. Tables and charts are legible with differences marked. There are quite few references, but the subject matter corresponds to the problem being discussed.

Author Response

Reviewer 4: Thanking you for your words, we will improve and explore possible control mechanisms as suggested.

Round 2

Reviewer 3 Report

Comments and Suggestions for Authors

The authors improved the revised ms it can be published.